# Suppression of external quantum efficiency rolloff in organic light emitting diodes by scavenging triplet excitons

Buddhika S. B. Karunathilaka [1], Umamahesh Balijapalli [1,2], Chathuranganie A. M. Senevirathne[1], Seiya Yoshida[1], Yu Esaki[1], Kenichi Goushi[1,2,3], Toshinori Matsushima[3,4 ✉], Atula S. D. Sandanayaka[1,3,5 ✉] & Chihaya Adachi [1,2,3,4 ✉]

Large external quantum efficiency rolloff at high current densities in organic light-emitting diodes (OLEDs) is frequently caused by the quenching of radiative singlet excitons by long-lived triplet excitons [singlet–triplet annihilation (STA)]. In this study, we adopted a triplet scavenging strategy to overcome the aforementioned STA issue. To construct a model system for the triplet scavenging, we selected 2,6-dicyano-1,1-diphenyl-$\lambda^5\sigma^4$-phosphinine (DCNP) as the emitter and 4,4′-bis[(N-carbazole)styryl]biphenyl (BSBCz) as the host material by considering their singlet and triplet energy levels. In this system, the DCNP's triplets are effectively scavenged by BSBCz while the DCNP's singlets are intact, resulting in the suppressed STA under electrical excitation. Therefore, OLEDs with a 1 wt.%-DCNP-doped BSBCz emitting layer demonstrated the greatly suppressed efficiency rolloff even at higher current densities. This finding favourably provides the advanced light-emitting performance for OLEDs and organic semiconductor laser diodes from the aspect of the suppressed efficiency rolloff.

---

[1] Center for Organic Photonics and Electronics Research (OPERA), Kyushu University, 744 Motooka, Nishi, Fukuoka 819-0395, Japan. [2] Education Center for Global Leaders in Molecular System for Devices, Kyushu University, 744 Motooka, Nishi, Fukuoka 819-0395, Japan. [3] Japan Science and Technology Agency (JST), ERATO, Adachi Molecular Exciton Engineering Project, 744 Motooka, Nishi, Fukuoka 819-0395, Japan. [4] International Institute for Carbon-Neutral Energy Research (WPI-I2CNER), Kyushu University, 744 Motooka, Nishi, Fukuoka 819-0395, Japan. [5] Department of Physical Sciences and Technologies, Faculty of Applied Sciences, Sabaragamuwa University of Sri Lanka, 70140 Belihuloya, Sri Lanka. ✉email: tmatusim@i2cner.kyushu-u.ac.jp; sandanay@appsc.sab.ac.lk; adachi@cstf.kyushu-u.ac.jp

Over the last few decades, a wide variety of materials and methods have been developed for improving performance of organic light-emitting diodes (OLEDs)[1–5]. Important advantages of OLEDs are easy fabrication, mechanical flexibility, light weight, emission colour tunability, low cost, and compactness[1–5]. One of the biggest issues frequently observed in OLEDs is that external quantum efficiencies (EQE) significantly decrease at high current densities which is termed as "efficiency rolloff". Major reason is the accumulation of long-lived triplet excitons, resulted in the singlet–triplet annihilation (STA)[6–9] and the decrease of singlet population density. On the other hand, if the charge balance in OLEDs is not perfect, charge accumulation at high current densities would occur, thereby leading to singlet–polaron annihilation (SPA)[10]. Recently, attention has been paid for research on laser devices based on organic materials[11–13]. Although laser oscillation from organic materials under optical pumping is comparatively easier[14], electrically pumped organic semiconductor laser diodes (OSLDs)[15] are still difficult to demonstrate due to so called triplet and polaron induced effects.

In working OLEDs and OSLDs, the singlet and triplet excitons are formed in a 1:3 ratio by the recombination of electrons and holes injected from electrodes. Thus, the current density is an extrinsic property that controls the rate of triplet exciton generation while the triplet lifetime is an intrinsic property of a molecule that controls the rate of triplet relaxation. Therefore, the triplet excitons start to accumulate when the rate of triplet formation by current, exceeds the rate of triplet relaxation of the molecules. Subsequently, the triplet accumulation reduces the population density of singlet excitons[16,17], thus hampering output electroluminescence (EL) intensity. Furthermore, some of triplet-exited-state molecules are unstable and, therefore easy to decompose, limiting the operational durability of OLEDs and OSLDs[18]. Thus, it is important to manage the triplet excitons to bring about the maximum performance from organic optoelectronic devices.

One way to suppress the STA is to use organic emitters, which have no spectral overlap between the singlet emission and the triplet absorption[8,15,19–21]. Other way to suppress the triplet accumulation is to use short-pulse excitation with longer relaxation time. Another possibility is the use of triplet scavenger molecules[22,23] such as oxygen[23,24], perylene derivatives[25], cyclooctatetraene[26,27] or anthracene derivatives[26,28]. To have efficient triplet scavenging, the triplet manager molecule should meet a set of important requirements[26]. The scavenger molecules should be able to accept the triplet excitons from the emitter molecules, which implies that the scavenger molecules should be close to the emitter molecules within the limit of bimolecular Dexter energy transfer (DET) distance and the triplet energy level ($T_1$) should be lower for the scavenger molecule than for the emitter molecules. At the same time, the singlet energy level ($S_1$) of the scavenger molecules should be high enough to prevent quenching of the singlet excitons of the emitter molecules. The scavenger molecules should possess a reasonably short intrinsic triplet lifetime to quickly deplete the triplet population so that triplet accumulation does not happen. The emission spectrum of the emitter molecules should be separated from the triplet absorption or polaron absorption of the scavenger molecules in order to suppress the annihilation processes between the emitter and scavenger molecules. Moreover, the scavenger molecules should not enhance intersystem crossing (ISC) of the emitter molecules which imposes certain limitations for employing compounds containing heavy atoms, such as metal–organic complexes, as triplet scavengers[26,29].

On the other hand, it is well considered that the $T_1$ level of the host molecules should be high enough for better triplet confinement especially, for second-generation phosphorescent OLEDs (PHOLEDs) and third-generation thermally activated delayed fluorescent (TADF) OLEDs[30–35]. However, first-generation OLEDs with the fluorescence emitters do not require the host molecules with a higher $T_1$ as the triplets do not contribute to EL in fluorescent OLEDs. Therefore, contrary to triplet confinement of conventional OLEDs, it is better to use the host molecules that have the triplet scavenging capability for itself, especially for fabricate OSLDs with fluorescence laser dyes. However, finding such materials, which fulfil the requirements for a host material as well as the requirements for a triplet scavenger material, are extremely limited[26].

In general, a good host material for OLEDs should possess the following set of important requirements[36–38]. The host molecules should have the emission spectrum overlapping with the absorption spectrum of the emitter molecules to induce efficient forward Förster resonance energy transfer (FRET). The host molecules should have bipolar charge transport properties for use in electrical operation. The host materials should be amorphous since the presence of grain boundaries impede carrier transport and scatter the light. The host molecules should not form degenerate energy states, which are associated with the formation of dimer or excimer states. Especially regarding the single-layer OLED architecture, the highest occupied molecular orbital (HOMO) and lowest unoccupied molecular orbital (LUMO) levels should be compatible with those of charge injection materials and the Fermi level of electrode materials in order to balance hole and electron currents. Moreover, the host molecules should not form an exciplex with emitter molecules to avoid the formation of any unwanted energy states.

Previously, we reported that an OLED architecture with a single neat emitting layer of BSBCz has well suppressed efficiency rolloff as compared to those of conventional multilayer architectures[9,15,39,40]. More recently, we showed an evidence of lasing action under current injection from a blue-emitting BSBCz-based OSLD with single-layer device architecture[15]. Thus, BSBCz fulfils all the aforementioned intrinsic requirements for the efficient host molecule. Previously, we reported true-CW laser operation using a guest–host matrix utilizing DCNP[41] as an active chromophore and BSBCz as the triplet scavenging host[42]. Therefore, with the intention to address the removal of triplets from the emitters under electrical excitation, thereby suppressing triplet induced issues, we introduced the same guest–host matrix into OLEDs in this work. Interestingly, this host-guest system showed excellent suppression of excited-state annihilation processes (i.e., STA) and efficiency rolloff at high current densities of electrical excitation. The results show a great prospect of using this guest–host system for future current injection OSLDs.

## Results

**Advantage of using a triplet scavenging host material**. In the conventional device architectures of PHOLEDs and TADF-OLEDs, the use of host materials that can confine triplet excitons is requisite for high efficiencies. Thus, in case of using 4,4′-bis(N-carbazolyl)-1,1′-biphenyl (CBP) as the host and DCNP as the emitter, both singlet and triplet exciton transfers occur from CBP to DCNP and those from DCNP to CBP are prevented (Fig. 1a), since both of singlet and triplet energy levels of CBP is higher than those of DCNP. The radiative decay constant of DCNP is $1.0 \times 10^8 \, s^{-1}$ [photoluminescence (PL) quantum yield = 0.83, fluorescence lifetime = 8.23 ns][42]. On the other hand, nonradiative decay rate of DCNP triplets is $2.9 \times 10^4 \, s^{-1}$ due to the long transient lifetime (35 μs)[42]. Therefore, DCNP emitters accumulate triplets at higher current densities, thereby suppressing the rate of singlet exciton formation and hampering the singlet emission.

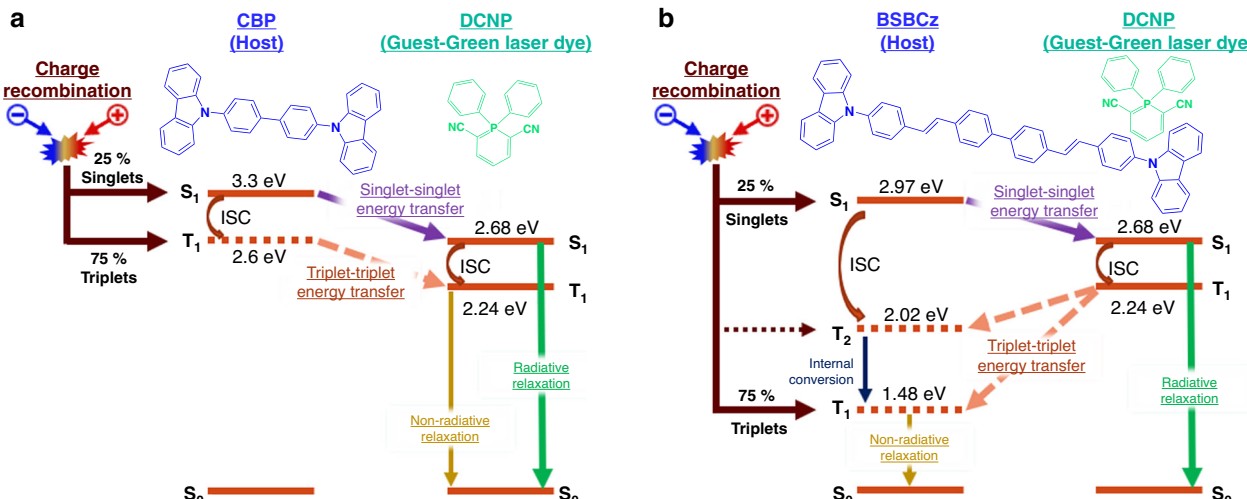

**Fig. 1 Energy level diagram of guest–host systems.** Energy transfer processes under electrical excitation when the guest material is DCNP and the host material is **a** CBP or **b** BSBCz. Chemical structure of each material is given with its abbreviated name. $S_0$, $S_1$, $T_1$, $T_2$, and ISC stand for the singlet ground state, the first singlet excited state, the first triplet excited state, the second triplet excited states, and the intersystem crossing, respectively. The triplet excited-state energies of BSBCz were taken from literature[18]. The other excited-state energies of CBP, DCNP, and BSBCz were estimated from the onset wavelengths of fluorescence spectra at room temperature and phosphorescence spectra at 77 K.

However, when BSBCz is employed as the host and the same molecule DCNP as the guest emitter, the triplet level of host molecules is lower than the triplet level of the guest molecule which is a different scenario explained above as with CBP as the host, thus it is impossible to transfer triplets from BSBCz host to DCNP guest molecules and thereby the triplets stay on the host BSBCz molecules themselves (Fig. 1b). In addition, whenever triplet excitons are formed on guest emitter DCNP molecules via ISC and direct exciton formation at the emitter molecules, then those triplet excitons will transfer to the BSBCz host molecules via DET mechanism. It is very easy to undergo this reverse DET because the molar ratio of host molecules is higher than that of guest molecules in the guest–host system. Therefore, it is obvious that the guest emitter molecules are always free to accept FRET from host's singlets and thereby emitter singlet exciton formation and relaxation cycles, i.e., singlet exciton formation frequency, is faster than the usual triplet associated case.

Moreover, the doping concentration has to be optimized in order to obtain minimum concentration quenching, and efficient FRET from host to guest singlets. Decreasing the doping concentration increases the distance between host and guest molecules and eventually decreases the FRET efficiency. Therefore, it is important to decrease the doping concentration until the marginal level of residual host emission can be observed. At this marginal concentration, we can get the maximum distance to suppress STA while keeping efficient FRET from host to guest. In this experiment, we used a 1 wt. %-DCNP:BSBCz system which corresponds to the molar ratio of 1:43 for DCNP:BSBCz. Even though, at this low doping ratio, the residual emission from host molecules was not observed, suggesting the very efficient FRET from host to guest as shown in Supplementary Fig. 1c. Favourably, using this low doping ratio, the triplet exciton formed on a neighbouring host molecule diffuses to another host molecule and migrates away from the emitter molecule, leading to the suppression of STA[43]. Moreover, triplet excitons are almost formed on host molecules and thereby avoiding the triplet induced degradation of guest emitter molecules which will ultimately lead to comparatively better device stability.

**Charge carrier balance in thin film diode.** Apart from the intrinsic properties of materials used for OLEDs, it is very important to consider about the device architecture in concern to other EQE governing factors of OLEDs. In conventional OLED architectures, multilayer interfaces play important role in balancing hole injection and electron injection which is one of the crucial factors for exciton formation efficiency, thereby it is directly related to the internal quantum yield. However, detrimentally there is a possibility that polaron absorption at the electron transport layers (ETL) and hole transport layers (HTL) may quench the light output[10,40], especially considering OSLDs where the light travels back and forth within the resonator structure for high optical feedback. Particularly, at high current densities, the polaron densities in ETLs and HTLs are very high, inducing rapid efficiency rolloff[44,45]. Therefore, using ETLs or HTLs will hamper the optical gain unless the polaron absorption spectra of both ETL and HTL materials do not overlap with laser emission spectra. Therefore, the easiest way to avoid this complex material selection for ETLs and HTLs is using a single-layer architecture of OLEDs as the first generation of OSLDs[15]. Nevertheless, in order to obtain high performance from the single-layer architecture, well-balanced hole and electron transport of an organic layer are strongly required. Hence, for the optimum performance of single-layer OLEDs, intrinsic properties of the host molecules play a very important role.

The HOMO levels of BSBCz and DCNP were $-5.9 \pm 0.1$ and $-6.0 \pm 0.1$ eV, respectively (Supplementary Fig. 1a), whereas LUMO of those were estimated to be $-3.1 \pm 0.1$ and $-3.5 \pm 0.1$ eV, respectively. To investigate the carrier transport properties of DCNP:BSBCz, we fabricated an electron only device (EOD) and a hole only device (HOD) with the structures of glass substrate/indium tin oxide (ITO) (30 nm)/20 wt.%-Cs:BSBCz (30 nm)/1 wt.%-DCNP:BSBCz (180 nm)/Cs (10 nm)/Al (100 nm) and glass substrate/ITO (30 nm)/MoO$_3$ (10 nm)/1 wt.%-DCNP:BSBCz (200 nm)/MoO$_3$ (10 nm)/Al (100 nm) (Fig. 2a). Current density–voltage curves of the EODs and HODs were measured under direct current (DC) and pulse operation. Notably, the current density–voltage curves of the EODs and HODs almost overlapped relative to each other in the whole voltage range

(Fig. 2b). This indicates that 1 wt.%-DCNP-doped BSBCz films have well-balanced electron and hole transport, leading to higher efficiency.

We believe that charge transport is mainly dominated by the host molecule of BSBCz, leading to similar charge balance as the reference device (Fig. 3, device B). In order to study whether the charge transport channel is affected by molecular orientation of BSBCz[46,47], the variable angle spectroscopic ellipsometry (VASE) measurements were conducted. The obtained $n$ and $k$ spectra are shown in Supplementary Fig. 2. Here, $k_x$ and $k_z$ values at a wavelength of 370.61 nm were used for the calculation of the orientation order parameter $S$ value, which corresponds to the π–π* transition of BSBCz. The molecular orientation discussed here

indicates the orientation of the transition dipole moment which is almost along the molecular long axis of BSBCz[48]. The calculated $S$ values of neat and doped films were −0.374 and −0.381, respectively, which are similar to our previous report[48] and it indicates that the DCNP doping does not change the molecular orientation in films. Although molecular orientation is one of the critical factor influencing charge carrier transport in organic semiconductor films[46–48], this effect is negligible in the comparison of our neat and doped films.

On the other hand, the HOMO and LUMO levels of CBP is reported to be −2.7 ± 0.1 and −6.1 ± 0.1 eV, respectively[49,50]. Anthopoulos et al. reported a highly efficient single-layer OLED using CBP as the host with the optimized thin charge injection layers[51]. In order to investigate the carrier transport properties of DCNP:CBP, we fabricated EODs and HODs with varying electron injection and hole injection layers as depicted in Supplementary Fig. 3a. Current density–voltage curves of the EODs and HODs were measured under DC operation as shown in Supplementary Fig. 3b, c. Since CBP is more favourable for hole transporting, attention has been paid to increase electron injection and decrease hole injection. With an increase of the Cs thickness and doping concentration, the amount of electron injection was increased (Supplementary Fig. 3). On the other hand, with the decrease of $MoO_3$ thickness, the amount of hole injection was increased. The EOD type C and HOD type A gave almost overlapped curves in the current density–voltage characteristics specially at high current density under DC operation (Supplementary Fig. 3), indicating the optimum charge balance in the OLED with the respective electron injection and hole injection layers.

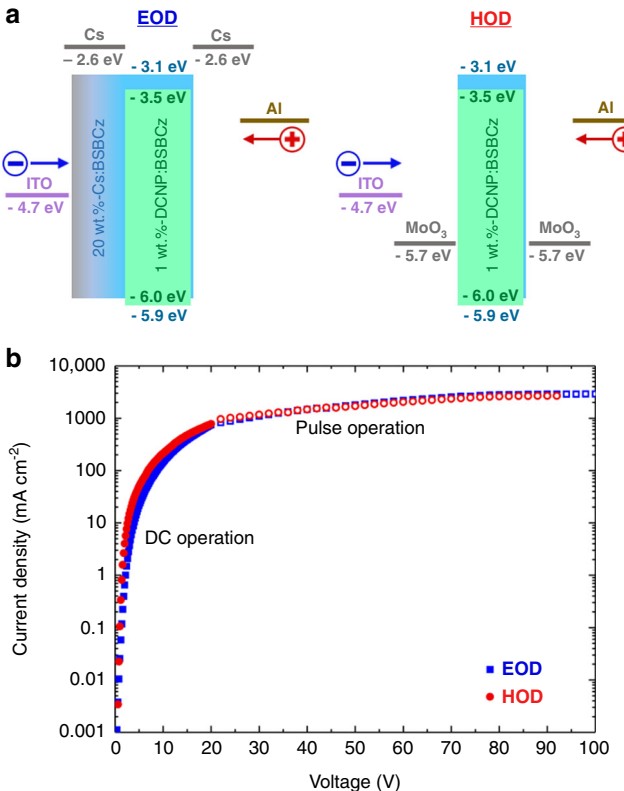

**Fig. 2 Evaluation of charge transport properties. a** Energy level diagrams for the EODs and HODs of single-layer devices. **b** current density–voltage ($J$-$V$) curves of EODs and HODs. Both hole current densities and electron current densities show perfect charge balance in the single-layer devices at both low voltage regime under DC operation and high voltage regime under pulsed operation.

**Electrical and electroluminescent properties.** Furthermore, we studied the EL properties of DCNP, separately using CBP and BSBCz as the host with single-layer OLEDs. Device A consists with a glass substrate/ITO (30 nm)/Cs (10 nm)/20 wt.%-Cs:CBP (30 nm)/1 wt.%-DCNP:CBP (180 nm)/$MoO_3$ (10 nm)/Al (100 nm), device B consists with a glass substrate/ITO (30 nm)/20 wt.%-Cs:BSBCz (30 nm)/BSBCz (180 nm)/$MoO_3$ (10 nm)/Al (100 nm) and device C consists with a glass substrate/ITO (30 nm)/20 wt.%-Cs:BSBCz (30 nm)/1 wt.%-DCNP:BSBCz (180 nm)/$MoO_3$ (10 nm)/Al (100 nm). The energy level diagrams of the OLEDs are shown in Fig. 3. Doping of Cs into BSBCz or CBP and the insertion of $MoO_3$ between organic layer and Al facilitate electron and hole injection from electrodes, respectively. Even though, the relatively shallower HOMO level of BSBCz and deeper LUMO level of DCNP may create an exciplex, we observed no exciplex emission because the EL spectrum is exactly similar as the DCNP PL spectrum in solution (Supplementary Fig. 1c). Moreover, considering the frontier orbital energy levels and the very low

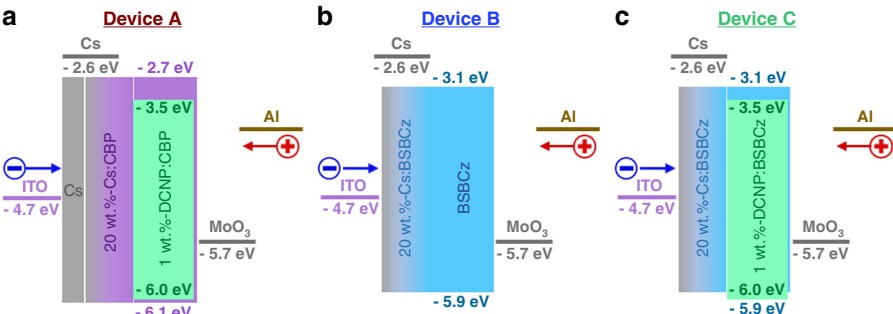

**Fig. 3 Energy level diagrams of the OLEDs.** EML consist with **a** 1 wt.%-DCNP:CBP (device A), **b** pristine BSBCz (device B), and **c** 1 wt.%-DCNP:BSBCz (device C) which were used to compare triplet accumulation property of CBP versus triplet scavenging property of BSBCz.

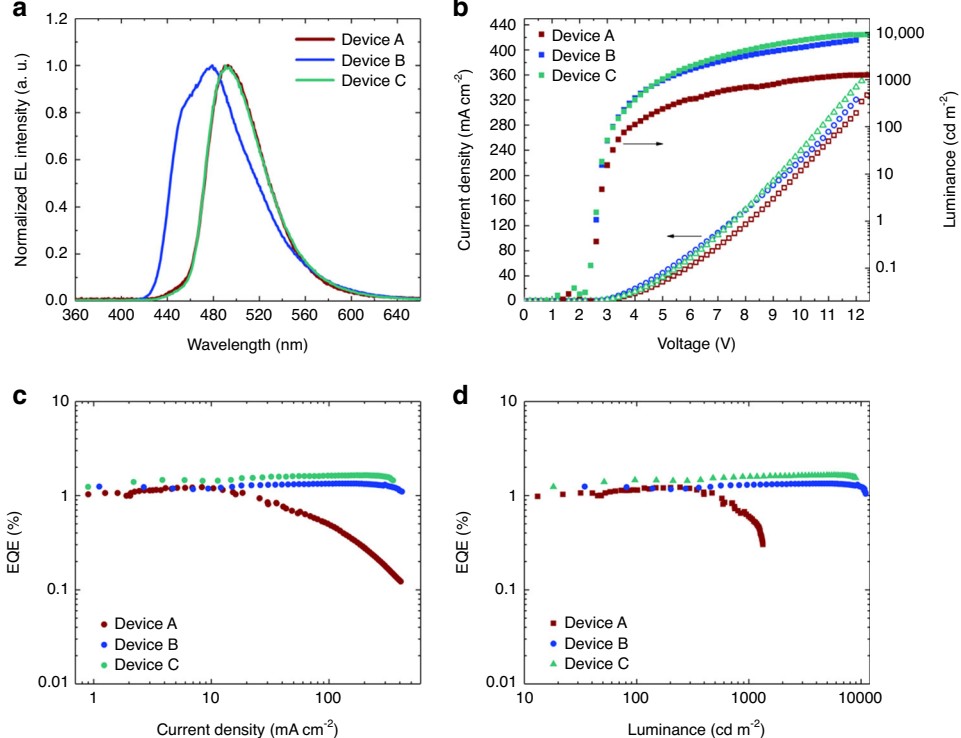

**Fig. 4 Comparison of OLED performance. a** EL spectra shows similar spectrums for devices A and C as the emission comes from DCNP emitter without significant shoulder peak of residual emission from host which suggests that efficient FRET from host to guest. **b** current density–voltage and luminance–voltage curves showing current density in all devices are approximately similar whereas significant decrement of luminance can be observed in device A due to severe STA. **c** EQE–current density curves showing serious EQE rolloff of device A at higher current densities comparing to devices B and C. **d** EQE–luminance curves showing maximum output luminance of device A is limited around 1200 cd m$^{-2}$ while EQE is dropping down due to STA, whereas devices B and C show maximum output luminance around 10,000 cd m$^{-2}$ with no serious EQE rolloff until the devices break down due to joule heat.

doping concentration of DCNP in BSBCz (1 wt.%), it would be reasonable that charge recombination mainly occurs on CBP or BSBCz host molecules. Thereafter, the formed singlet exciton energy will be transferred into DCNP via FRET. As shown in Fig. 4a and Supplementary Fig. 1c, both DCNP-doped devices showed similar emission spectra to the PL of DCNP without yielding additional shoulder peaks from residual host emission, which suggests almost perfect FRET from host to guest.

In the case of CBP host device (device A), the luminance output is lower (Fig. 4b) and efficiency rolloff is higher at larger current densities (Fig. 4c) due to accumulation of triplet excitons which causes STA. In the case of BSBCz host devices (device C), triplet excitons formed on BSBCz molecules would not transferrable and hence it would be relaxed via nonradiative relaxation pathways as clarified earlier in Fig. 1b. Faster nonradiative relaxation of BSBCz triplet excitons was already reported, as the triplet exciton lifetime is in nano-second range[42,52]. Advantageously, due to suppressed STA in BSBCz-based devices (devices B, C), efficiency rolloff suppressed until the current density is 500 mA cm$^{-2}$, where after the device starts to breakdown due to joule heating under DC operation (Fig. 4c, d). EQE of an OLED can be expressed as

$$EQE \% = \gamma \times \beta \times \phi \times \eta \times 100\% \tag{1}$$

where $\gamma$ = charge recombination efficiency, $\beta$ = exciton formation probability, $\phi$ = PL quantum yield, $\eta$ = light outcoupling efficiency. Based on the results of EOD and HOD, well-balanced electron and hole transport is expected in all devices, hence, $\gamma$ is expected to be closer to 1. In 1 wt.%-DCNP:BSBCz host-guest matrix, PLQY is 0.83 and singlet exciton formation probability

should be 0.25 according to spin statistics under electrical excitation. Therefore, if light outcoupling from the device is assumed to be 0.2, then the device should exhibit maximum external quantum efficiency (EQE$_{max}$) of 4.15% based on Equation 1. However, in the experimental conditions, it shows (1.4 ± 0.1%), (1.4 ± 0.1%), and (1.5 ± 0.1%) EQE$_{max}$ for device A, B, and C, respectively (Fig. 4c). Detrimentally, the single-layer devices do not have an electron blocking layer or a hole blocking layer to confine charge recombination in the emissive layer which is typical in multilayer device architecture. Therefore, some of the electrons and holes could reach counter electrodes without recombination, leading to low efficiency in the single-layer devices, although balanced carrier mobilities were observed. However, advantageously for OSLDs, this single-layer device architecture is very effective for suppressing the efficiency-rolloff, due to SPA especially at higher current densities[15].

In order to demonstrate the universality of this triplet scavenging guest–host matrix, we selected another common emitter molecule, coumarin 545T (C545T). Even though, any emitter molecules similar to DCNP should show similar triplet scavenging behaviour, it is important to note that the materials should satisfy no spectral overlap between the singlet emission of guest molecules and the triplet excited-state absorption of host molecules to escape the annihilation processes between them, i.e., emitter–host STA. As shown in Supplementary Figs. 4 and 5, C545T has good agreement of singlet energy transfer from host to guest and triplet energy transfer from guest to host. Therefore, an OLED was fabricated with the emissive layer (EML) consisted of 1 wt.%-C545T:BSBCz as depicted Supplementary Fig. 6. As expected, the device with coumarin 545T as the emitter showed

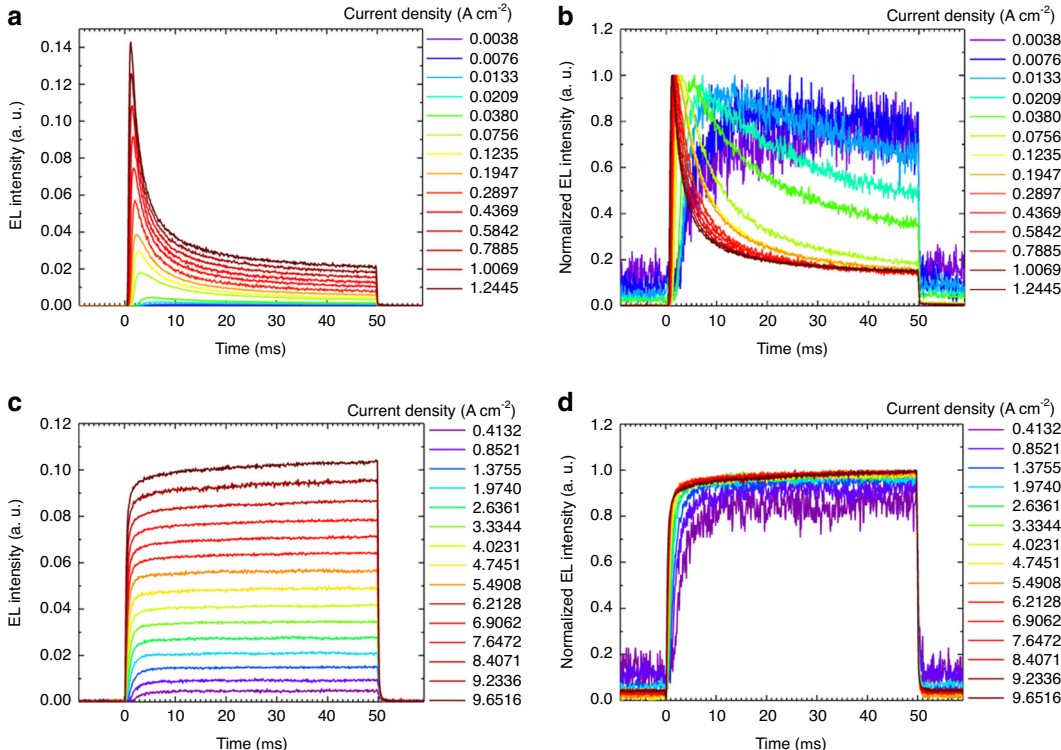

**Fig. 5 Transient EL responses of OLEDs. a**, **b** When the host is CBP for device A or **c**, **d** when the host is BSBCz for device C, which were measured at different current densities with a pulse width of 50 μs. For (**b**) and (**d**) respectively, the EL intensities were normalized to make it easy to see the response shapes. The EL intensity from OLEDs were recorded using a photomultiplier tube (PMT) (C9525-02, Hamamatsu Photonics) for the measurement of temporal emission profile.

similar behaviours such as suppression of efficiency rolloff at high current densities (Supplementary Fig. 7), indicating the universality of using BSBCz as a triplet scavenging host.

**Suppressed EL quenching**. The circuit diagram used for transient EL measurement is depicted in Supplementary Fig. 8. As shown in Fig. 5a, b, when DCNP was doped in a CBP host, it shows clear quenching of singlet emission (EL intensities) by long-lived triplet excitons with time when OLEDs are driven at 50 μs square pulses. At low current densities, EL was rising because the triplet induced EL quenching was not in progress (Fig. 5b). However, when the devices are operated at higher current densities, the accumulation of triplets undergoes annihilation processes with singlet excitons, thereby decreasing EL intensity by time.

On the other hand, very interestingly, our devices with 1 wt. %-DCNP:BSBCz showed complete suppression of EL-STA when driven at 50 μs square pulses even until 10 A cm$^{-2}$ very high current density (Fig. 5c, d). This is evidenced by no decrease in EL intensity. Major reason is that triplet scavenging mechanism realized using BSBCz as the host instead of CBP, led to the complete removal of triplets from the emitter molecules, thereby completely avoiding the emitter–emitter STA. The other reasons are negligible overlap of the DCNP emission spectrum with the BSBCz triplet absorption spectrum[42], while faster triplet relaxation lifetime of BSBCz became an additional advantage[42,52] to complete suppression of emitter–host STA.

In order to investigate the effect of charge accumulation and thereby the effect of charge induced emission quenching, both current and EL intensity responses were compared over the pulse width of 50 μs (Supplementary Fig. 9). As shown in Supplementary Fig. 9, the injected current stays unchanged while EL intensity is gradually decreased, indicating charge

accumulation is not significant, and thus charge-induced exciton quenching is not severe, but triplet exciton induced quenching would be significant.

In addition, as shown in Supplementary Fig. 10, we observed no off-state EL which is a typical observation when there is considerable charge accumulation in OLEDs[10]. Supplementary Fig. 10b, d showing the expanded parts at the end of transient EL profiles of both CBP host (Device A) and BSBCz host (Device C) OLEDs clearly show no such off-state EL. This result provides additional evidence that charge balance in all devices is ideal and no serious charge accumulation and therefore no SPA occurred in the devices.

In order to find out triplet accumulation at longer pulse width, we tested transient EL at 100, 200, and 500 μs and even up to 1 ms square pulses driven until 10 A cm$^{-2}$ current density (Fig. 6). Surprisingly, it showed no EL quenching even at 1 ms long pulse width and 10 A cm$^{-2}$ very high current density (Fig. 6g, h). This result is highly promising for prospective OSLD devices in order to reach high optical gain at higher current densities and for better device stability.

## Discussion

Objective of the present invention is to demonstrate the effectiveness of a triplet scavenging guest–host matrix for suppressing EL-STA and efficiency rolloff at higher current densities of OLEDs. Triplet scavenging mechanism was successfully realized by using DCNP as a fluorescent emitter and BSBCz as the host in a guest–host matrix. Doping DCNP into BSBCz yielded similar EQE as non-doped devices. It showed clear STA even at 50 μs pulsed voltage when CBP was used as the host, while on the other hand using BSBCz as the host showed no EL quenching by STA, SPA or any other manner from 50 μs to 1 ms long pulse width

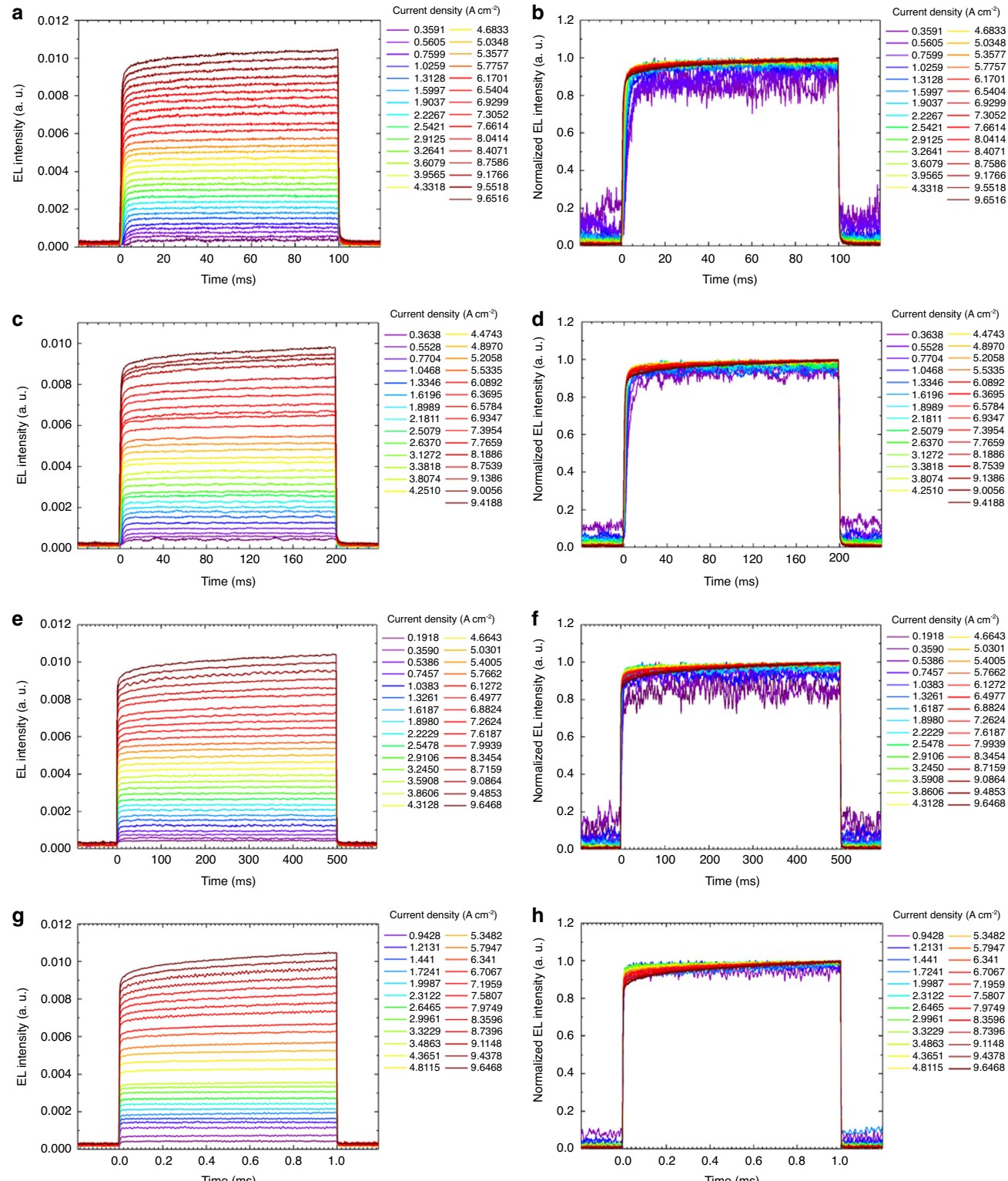

**Fig. 6 Transient EL responses of OLED device C at long pulses.** Temporal profiles of EL intensity when the pulse widths were **a**, **b** 100 µs, **c**, **d** 200 µs, **e**, **f** 500 µs, and **g**, **h** 1 ms. For (**b**, **d**, **f**, and **h**), the EL intensities were normalized to make it easy to see the response shapes. The EL intensity from OLEDs were recorded using a photomultiplier tube (PMT) (C9525-02, Hamamatsu Photonics) for the measurement of temporal emission profile.

until the very high current density of 10 A cm$^{-2}$. The suppressed STA and suppressed efficiency rolloff for DCNP:BSBCz-based OLED devices showed an interesting material combination and device architecture for the fabrication of prospect OSLDs with high performance.

## Methods

**Estimation of frontier orbital energy levels.** To estimate the energy level of HOMO of all materials, 100 nm thin films of pristine BSBCz, DCNP, and 1 wt. %-DCNP:BSBCz were vacuum-deposited on pre-cleaned ITO-coated glass substrates. The HOMO energy levels were determined using photoelectron yield spectroscopy (AC-3, Riken-Keiki) in neat films, and then the LUMO energy levels

were estimated by subtracting the optical energy gap ($E_g$) from the measured HOMO energies. In the case of the $E_g$ values were determined from the onset of the PL spectra of neat films.

**Single-carrier devices and OLED fabrication**. To fabricate EOD, HOD, and OLEDs, all organic and metal layers were vacuum-deposited on clean pattern ITO-coated glass substrates under a pressure of $1 \times 10^{-4}$ Pa. The deposition rates were 1.0 Å s$^{-1}$ for Cs doped films of BSBCz and CBP, 2.5 Å s$^{-1}$ for DCNP-doped films of BSBCz or CBP, 0.3 Å s$^{-1}$ for MoO$_3$, and 1.0 Å s$^{-1}$ for Al. The completed OLEDs were directly transferred into a nitrogen filled glovebox and encapsulated using a glass cap and an UV curing epoxy resin.

**Single-carrier devices and OLED characterization**. The current density–voltage ($J$–$V$), external quantum efficiency–current density curves, and EL spectra at DC operation were measured using an external quantum efficiency measurement system (C9920-12, Hamamatsu Photonics). Pulsed voltage operation was measured using rectangular pulses with a pulse width of 400 ns, repetition frequency of 1 kHz, and varying peak voltages were applied to the devices at ambient temperature using a pulse generator (NF, WF1945) while applied voltage ($V_{CH1}$) was monitored on a multichannel oscilloscope (MSO6104A, Agilent Technologies). To measure current flow through the device a 51.4 Ω resistor was connected in serial connection to the OLED and voltage across the resistor ($V_{CH2}$) was monitored on the oscilloscope (Supplementary Fig. 8). Therefore, in situ voltage across the OLED device can be reported as $V_{CH1}$–$V_{CH2}$ based on voltage division rule (Supplementary Fig. 8).

**VASE measurement**. In order to estimate a molecular orientation of BSBCz, spectra of refractive index $n$ and extinction coefficient $k$ of vacuum-deposited neat and 1 wt.% DCNP-doped films of BSBCz were obtained using ellipsometer (VASE; M-2000, J.A. Woollam). The measurement was performed on the 100-nm-thick films deposited on bare Si substrates. The incident angle was varied from 45° to 75° with a 5° step increment and the spectral measurement range was set to 245–1000 nm. BSBCz films are known to show anisotropy along the directions parallel and perpendicular to the substrate plane because of the orientation of molecules. Therefore, VASE results of BSBCz films were fitted using an uniaxial anisotropic optical model considering molecular anisotropy, and the orientation order parameter $S$ was calculated using following Eq. (2)[46–48];

$$S = \frac{k_z - k_x}{k_z + 2k_x} \quad (2)$$

where $k_x$ and $k_z$ are the extinction coefficients in the directions parallel and perpendicular to the substrate, respectively. It should be noted that the same optical model was used for neat and doped films, assuming that the doping concentration of DCNP is sufficiently small (1 wt.%).

**Transient EL at long-pulsed voltage**. Long-pulsed voltage was generated same as the short-pulsed voltage measurement explained in Single-carrier devices and OLED characterization and Supplementary Fig. 8. The EL intensity from OLED was recorded using a photomultiplier tube (PMT) (C9525-02, Hamamatsu Photonics) for the measurement of temporal emission profile. Driving square-wave voltage signal ($V_{CH1}$), voltage across resistor ($V_{CH2}$), and PMT response ($V_{CH3}$) were monitored on a multichannel oscilloscope (MSO6104A, Agilent Technologies). The circuit diagram and schematic of BNC cable connection are depicted in Supplementary Fig. 8.

## Data availability

The data that support the findings of this study are available from the corresponding author upon reasonable request.

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

## Acknowledgements

We thank Dr. Takashi Fujihara, Dr. Jun-ichi Nishide, and Ms. Shinobu Terakawa for their technical assistance. This work was supported by the Japan Society for the Promotion of Science (JSPS) core to core programme and the Ministry of Education, Culture, Sports, Science, and Technology (MEXT), Japan.

## Author contributions

B.S.B.K., A.S.D.S., T.M., and C.A. conceived the project. B.S.B.K. designed and conducted the experiments, analysed the data and wrote the manuscript. U.B. synthesized the DCNP molecules in larger scale for vacuum thermal evaporation. C.A.M.S. and S.Y. contributed for transient EL measurement. Y.E. contributed for the VASE measurement. K.G., T.M., A.S.D.S., and C.A. revised the manuscript. All authors discussed the results and commented on the manuscript.

## Competing interests

The authors declare no competing interests.
