## [Peer Review File · Nature Communications]

Reviewers' Comments:

Reviewer #2:

Remarks to the Author:

The authors have studied the triplet scavenging strategy to overcome the singlet-triplet annihilation (STA) issue. Usually the high EQE roll off will cause by STA. In this paper they have studied the triplet scavenging strategy by considering the 2,6-dicyano-1,1-diphenyl- λ 5 σ 4-phosphinine (DCNP) as the emitter and 4,4'-bis[(N-carbazole)styryl] biphenyl (BSBCz) as the triplet scavenging host material in order to get suppressed STA. They observed OLEDs with 1wt% DCNP doped BSBCz emitting layer shows suppressed EQE roll off at large current densities. The similar work already done by the same group with ter(9,9'-spirobifluorene) (TSBF) doped in a host matrix layer of 4,4'-bis(carbazol-9-yl)biphenyl (CBP) (Appl. Phys. Lett. 108, 133302 (2016)). The work looks interesting and catching attention for the future OLED technology. By considering the literature survey and data information provided, the quality of this manuscript is suitable to this journal. So I recommend for acceptance of this manuscript. However, for further consideration, I recommend the authors to consider the following comments.

1. In introduction, kindly clear cut the why efficiency roll off occurs?
2. In explanation of good host molecule requirements point (6) is not clear.
3. In results section, Mention about different host BSBCz in figure1b matter (para on page no7).
4. Explain some more clear about how FRET process type of low doping ratio could help to suppress STA?
5. In case of figure2, kindly keep the HOD first and then EOD. Because the authors mentioned the order HOD first and then EOD in page no8.
6. In discussion authors mentioned λ 5-phosphinine fluorescent emitter. Kindly correct it.
7. The authors did not mention about a low amplified spontaneous emission (ASE) threshold. Why?
8. The authors did not provide the radiative decay constant values and orientation factor. Kindly check and clear it.

Reviewer #3:

Remarks to the Author:

This paper reports a triplet scavenging strategy to suppress the singlet-triplet annihilation (STA) for reducing the quenching of radiative singlet excitons induced by long-lived triplet excitons. In this paper, the BSBCz and DCNP are used to form a host-guest system to prevent the energy transfer to the triplet states of DCNP due to the low triplet energy of the BSBCz host. As a result, the efficiency roll-off is significantly suppressed at high current densities. The model for the triplet scavenging is rather interesting, and the authors provide a comprehensive analysis of the material strategy for both host and emitter in the host-guest systems. However, the authors are suggested in addressing some issues before this paper can be recommended for publication.

1. In this paper, the DCNP was selected as an emitter due to its suitable singlet and triplet energies as compared to those of BSBCz. The energy transfer from the BSBCz's triplet to that of DCNP's triplet state is impossible, which suppress the possibility of STAs at high current density. However, the BSBCz is well known about its extremely low efficiency roll-off at high current density, which favors the possibility of the electrical lasing as demonstrated by the same group of this paper. Therefore, it would be more convincing to demonstrate the same triplet scavenging behavior by using one more material as an emitter in the BSBCz host, which will be important to rule out the contribution of the stable singlet energy transfer from BSBCz to the dopant emitter instead of the triplet scavenging and STA-related loss.
2. In this paper, the commonly used CBP is selected as a reference to that of BSBCz. However, the CBP has no good bipolar charge transport properties. It is believed that the significant efficiency roll-off will occur at high current density due to the unbalanced electron and hole transport when the CBP is used as a host for the dopant emission. On page 10, the efficiency roll-off at high current density for CBP host device (Device A) is attributed to the accumulation of triplet excitons and the induced STA. How about the influence of the unbalanced carrier transport and the possible

charge-induced exciton quenching?

3. In Figure 2, the electron-only and hole-only devices were measured for the DCNP:BSBCz-based single layer devices. It will be useful to compare the difference of unbalanced hole and electron transport for the DCNP:CBP devices.

4. The transient EL measurements of two host-guest systems were performed to compared the transient EL performances as shown in Figure 4. The authors conclude that the EL intensity drop of the CBP-based device (Device A) is due to the STA effect. However, it is known that the CBP has no good bipolar transport properties. How to exclude the charge-induced quenching effect at high current densities? In addition, it is not clear what is the wavelength for the transient EL measurements in Figure 4 and Supplementary Figure S4.

5. There are some spelling and grammar errors in the text, which should be corrected.

Comments and response (Reviewer #2)

Comment - Remarks to the Author:

The authors have studied the triplet scavenging strategy to overcome the singlet-triplet annihilation (STA) issue. Usually the high EQE roll off will cause by STA. In this paper they have studied the triplet scavenging strategy by considering the 2,6-dicyano-1,1-diphenyl- $\lambda^5\sigma^4$ -phosphinine (DCNP) as the emitter and 4,4'-bis[(*N*-carbazole)styryl] biphenyl (BSBCz) as the triplet scavenging host material in order to get suppressed STA. They observed OLEDs with 1wt% DCNP doped BSBCz emitting layer shows suppressed EQE roll off at large current densities. The similar work already done by the same group with ter(9,9'-spirobifluorene) (TSBF) doped in a host matrix layer of 4,4'-bis(carbazol-9-yl)biphenyl (CBP) (Appl. Phys. Lett. 108, 133302 (2016)). The work looks interesting and catching attention for the future OLED technology. By considering the literature survey and data information provided, the quality of this manuscript is suitable to this journal. So, I recommend for acceptance of this manuscript. However, for further consideration, I recommend the authors to consider the following comments.

Reply:

We thank you very much for your careful reading and positive assessment to our manuscript.

Comment 1:

1. In introduction, kindly clear cut the why efficiency roll off occurs?

Response:

We included the concise explanation of the efficiency rolloff, STA and SPA, in OLEDs as follows.

Revisions:

The major reason for the efficiency rolloff originates from the accumulation of long-lived triplet excitons. As a result, serious quenching of radiative singlet excitons by triplet excitons [singlet-triplet annihilation (STA)]⁴⁻⁷ and decreasing the singlet population density occurs. On the other hand, unless charge balance in the OLEDs is not perfect, charge accumulation at high current densities would occur, thereby leading to singlet-polaron annihilation (SPA)⁸.

Comment 2:

2. In explanation of good host molecule requirements point (6) is not clear.

Response:

We think that a good host should satisfy the following conditions and we included the detailed explanation by revising point 6) as point 5-b) (page 6) as follows:

Revisions:

5) It should have a good matching of HOMO and LUMO levels with those of emitter molecules and electrodes.

5-a) It should not form an exciplex with emitter molecules so as to avoid the formation of any unwanted energy states.

5-b) Especially regarding the single-layer OLED architecture, the HOMO and LUMO levels should be compatible with those of charge injection materials and the Fermi level of electrode materials in order to balance hole and electron currents.

Comment 3:

3. In results section, Mention about different host BSBCz in figure1b matter (para on page no7).

Response:

In addition to BSBCz as the host material, we also compared the triplet scavenging effect by using a common host material of CBP to obtain the insight about the function of BSBCz that efficiently works as a triplet scavenger. We clarified that CBP has no effect to scavenge triplets from DCNP while BSBCz can effectively scavenge triplets from DCNP. Thus, we included the more detailed explanation in the main text (page 7) as follows.

Revisions:

In the conventional device architectures of PHOLEDs and TADF-OLEDs, the use of host materials that can confine triplet excitons is requisite for high efficiencies. Thus, in case of using 4,4'-bis(*N*-carbazolyl)-1,1'-biphenyl (CBP) as the host and DCNP as the emitter, both singlet and triplet exciton transfers occur from CBP to DCNP and those from DCNP to CBP are prevented (**Figure 1a**), since both of singlet and triplet energy levels of CBP is higher than those of DCNP. Therefore, DCNP emitters accumulate triplets at higher current densities, thereby suppressing the rate of singlet exciton formation and hampering the singlet emission.

Comment 4:

4. Explain some more clear about how FRET process type of low doping ratio could help to suppress STA?

Response:

We included the detailed explanation on it in page 8 as follows.

Revisions:

Moreover, the doping concentration has to be optimized in order to obtain minimum concentration quenching, and efficient FRET from host to guest singlets. Decreasing the doping concentration increases the distance between host and guest molecules and eventually the FRET process decreases. Therefore, it is important to decrease the doping concentration until the marginal level of residual host emission can be observed. At this marginal concentration, we can get the maximum distance to suppress STA while keeping efficient FRET from host to guest.

Favourably, using this low doping ratio, the triplet exciton formed on a host molecule diffuses to another host molecule and migrates away from the emitter molecule, leading to the suppression of STA⁴¹.

Comment 5:

5. In case of figure2, kindly keep the HOD first and then EOD. Because the authors mentioned the order HOD first and then EOD in page no8.

Response:

As suggested, we rearranged the order of EOD and HOD in alphabetical order throughout the manuscript.

Comment 6:

6. In discussion authors mentioned λ^5 -phosphinine fluorescent emitter. Kindly correct it.

Response:

As suggested, in order to avoid confusion, we replaced the word " λ^5 -Phosphinine" into "DCNP". In addition, the following reference paper was added into the introduction.

Revisions:

Ref. 39: Hashimoto, N. *et al.* Synthesis and Photophysical Properties of λ^5 -Phosphinines as a Tunable Fluorophore. *J. Am. Chem. Soc.* **140**, 2046–2049 (2018).

Comment 7:

7. The authors did not mention about a low amplified spontaneous emission (ASE) threshold. Why?

Response:

Since we published about the light amplification characteristics in a separate paper, we did not include the result. Thus, we referred our very recent publication about the ASE characteristics. Here, we shortly summarize the ASE characteristics under optical excitation for reviewer's understanding.

The 1 wt.-%-DCNP:BSBCz guest-host matrix demonstrated a low ASE threshold of $1.1 \mu\text{J cm}^{-2}$ (Figure R1a,b). With an increase of DCNP doping concentrations, the PLQYs decreased while the ASE threshold increased accordingly (Figure R1b) and ASE peaks were redshifted (Figure R1c).

Figure R1. ASE from a DCNP:BSBCz film on a quartz substrate in nitrogen environment; (a) Emission intensity versus excitation intensity plot of 1 wt.-%-DCNP:BSBCz film, (b-inset) Photograph of ASE operation at 5 times above threshold, (b) Average ASE threshold and PLQY versus DCNP doping concentration in BSBCz. (c) Normalized ASE spectra variation upon DCNP doping concentration in BSBCz.

Revisions:

Ref. 40: Karunathilaka, B. S. B. *et al.* An Organic Laser Dye having a Small Singlet-Triplet Energy Gap Makes the Selection of a Host Material Easier. *Adv. Funct. Mater.* (in press) DOI: 10.1002/adfm.202001078

Comment 8:

8. The authors did not provide the radiative decay constant values and orientation factor.

Kindly check and clear it.

Response:

The orientation order parameters (S) of both doped and neat films are summarized in the results section (page 9) and the material and method section (page 14). We also summarized the result of the radiative decay constant in the manuscript with the corresponding reference #40 (page 7).

Revisions:

The radiative decay constant of DCNP is $1.0 \times 10^8 \text{ s}^{-1}$ (PLQY=0.83, $\tau_{\text{FL}}=8.23 \text{ ns}$)⁴⁰. On the other hand, non-radiative decay rate of DCNP triplets is $2.9 \times 10^4 \text{ s}^{-1}$ due to the long transient lifetime (35 μs)⁴⁰. Therefore, DCNP emitters accumulate triplets at higher current densities, thereby suppressing the rate of singlet exciton formation and hampering the singlet emission.

Ref. 40: Karunathilaka, B. S. B. *et al.* An Organic Laser Dye having a Small Singlet-Triplet Energy Gap Makes the Selection of a Host Material Easier. *Adv. Funct. Mater.* (in press). DOI: 10.1002/adfm.202001078

Comments and response (Reviewer #3)

Comment - Remarks to the Author:

This paper reports a triplet scavenging strategy to suppress the singlet–triplet annihilation (STA) for reducing the quenching of radiative singlet excitons induced by long-lived triplet excitons. In this paper, the BSBCz and DCNP are used to form a host-guest system to prevent the energy transfer to the triplet states of DCNP due to the low triplet energy of the BSBCz host. As a result, the efficiency roll-off is significantly suppressed at high current densities. The model for the triplet scavenging is rather interesting, and the authors provide a comprehensive analysis of the material strategy for both host and emitter in the host-guest systems. However, the authors are suggested in addressing some issues before this paper can be recommended for publication.

Reply:

We thank you for your careful reading and the positive evaluation to our manuscript.

Comment 1:

1. In this paper, the DCNP was selected as an emitter due to its suitable singlet and triplet energies as compared to those of BSBCz. The energy transfer from the BSBCz's triplet to

that of DCNP's triplet state is impossible, which suppress the possibility of STAs at high current density. However, the BSBCz is well known about its extremely low efficiency roll-off at high current density, which favours the possibility of the electrical lasing as demonstrated by the same group of this paper. Therefore, it would be more convincing to demonstrate the same triplet scavenging behaviour by using one more material as an emitter in the BSBCz host, which will be important to rule out the contribution of the stable singlet energy transfer from BSBCz to the dopant emitter instead of the triplet scavenging and STA-related loss.

Response:

As suggest, the universality of this system is important. Basically, any emitter molecules similar to DCNP should show similar triplet scavenging behavior. However, there is a limitation for the combination of guest and host materials. The materials should satisfy no spectral overlap between the singlet emission of guest molecules and the triplet excited state absorption of host molecules to escape the annihilation processes between them. Here, we selected another common emitter molecule, coumarin545T and observed the similar behaviors.

Revisions:

In order to demonstrate the universality of this triplet scavenging guest-host matrix, we selected another common emitter molecule, coumarin545T. Even though, any emitter molecules similar to DCNP should show similar triplet scavenging behaviour, it is important to note that the materials should satisfy no spectral overlap between the singlet emission of guest molecules and the triplet excited state absorption of host molecules to escape the annihilation processes between them, i.e., emitter-host STA. As shown in Supplementary Figures S4 and S5, coumarin 545T has good agreement of singlet energy transfer from host to guest and triplet energy transfer from guest to host. Therefore, an OLED was fabricated with the EML consisted of 1 wt.%C545T: BSBCz as depicted Supplementary Figure S6. As expected, the device with coumarin 545T as the emitter showed similar behaviours such as suppression of efficiency rolloff at high current densities (Supplementary Figure S7), indicating the universality of using BSBCz as a triplet scavenging host.

Supplementary Figure S4. Absorption and fluorescence emission spectra at room temperature and phosphorescence emission spectra at 77 K of coumarin 545T in solutions and solid-state films.

Supplementary Figure S5. Jablonski diagram showing energy transfer in a C545T:BSBCz layer.

Supplementary Figure S6. Energy level diagrams for the single layer OLED having C545T as the emitter and BSBCz as the host.

Supplementary Figure S7. OLED characteristics with C545T as the emitter and BSBCz as the host. (a) The EL spectrum is similar to the PL of C545T with no extra shoulder peaks, suggesting the efficient FRET from host to guest. (b) Current density–voltage and luminance–voltage curves showing similar results as Device C (c)

The EQE–current density curve showing no serious EQE rolloff until the device break-down due to joule heating. (e) The EQE–luminance curve showing the maximum output luminance over 10000 cd m^{-2} with no rolloff of EQE until the device break-down.

Comment 2:

2. In this paper, the commonly used CBP is selected as a reference to that of BSBCz. However, the CBP has no good bipolar charge transport properties. It is believed that the significant efficiency roll-off will occur at high current density due to the unbalanced electron and hole transport when the CBP is used as a host for the dopant emission. On page 10, the efficiency roll-off at high current density for CBP host device (Device A) is attributed to the accumulation of triplet excitons and the induced STA. How about the influence of the unbalanced carrier transport and the possible charge-induced exciton quenching?

Response:

Before we fabricate all devices, we carefully optimized the charge balance in all devices by controlling charge injection layers. As suggested, it is essential to realize charge balance in the CBP based device. Thus, we added the following explanation in page 10 and figures in supplementary section (Supplementary Fig. S3).

Revisions:

On the other hand, the HOMO and LUMO levels of CBP is reported to be -2.7 ± 0.1 and -6.1 ± 0.1 eV, respectively^{47,48}. T. D. Anthopoulos *et al.* reported a highly efficient single-layer OLED using CBP as the host with the optimized thin charge injection layers⁴⁹. In order to investigate the carrier transport properties of DCNP:CBP, we fabricated EODs and HODs with varying electron injection and hole injection layers as depicted in Supplementary Figure S3a. Current density–voltage curves of the EODs and HODs were measured under direct current (DC) as shown in Supplementary Figure S3b,c. Since CBP is more favorable for hole transporting, attention has been paid to increase electron injection and decrease hole injection. With an increase of the Cs thickness and doping concentration, the amount of electron injection was also increased (Supplementary Figure S3). On the other hand, with the decrease of MoO₃ thickness, the amount of hole injection was increased. The EOD type C and HOD type A gave almost overlapped curves in the current density–voltage characteristics specially at high current density under DC operation (Supplementary Figure S3), indicating the optimum charge balance in the OLED with the respective electron injection and hole injection layers.

Supplementary Figure S3. Optimizing charge balance in a single layer OLED with CBP as the host using EODs and HODs. (a) Device architecture of EODs and HODs with a 1wt.% DCNP:BSBCz single layer and varying electron injection and hole injection layers. Current density–voltage (J–V) curves of EODs and HODs; (b) Linear plot and (c) Semi-log plot.

From the comparison of EOD type A, B and C, the combination of 10 nm thick-Cs and Cs doped layer enhanced electron injection efficiency, while the decreased MoO₃ thickness enhanced hole injection efficiency. Based on these results, we found that the combination of EOD type C and HOD type A provides the optimum charge balance specially at high current density under DC operation.

Further, under pulse voltage operation, we compared the current density response curve with the EL response curve of Device A to clarify the effect of charge-induced exciton quenching. As shown in Supplementary Figure S5, after the capacitance current flow (until 10 μs), the injected current stayed unchanged. This means that the EL quenching occurs not due to the charge accumulation in Device A, but due to the triplet accumulation on DCNP emitter molecules. Following explanation and figures were added into the manuscript (page 13) and supplementary section (Supplementary Figure S9).

Revision:

In order to investigate the effect of charge accumulation and thereby the effect of charge induced emission quenching, both current and EL intensity responses were compared over the pulse width of 50 μs (Supplementary Figure S9). As shown in Supplementary Figure S9, the injected current stays unchanged while EL intensity is gradually decreased, indicating charge accumulation is not significant, and thus charge-induced exciton quenching is not severe, but triplet exciton induced quenching would be significant.

Supplementary Figure S9. Transient EL responses when the host is CBP for device

A. (a) Current density response over pulse width of 50 μs , (b) EL intensity response over pulse width of 50 μs . The capacitance current flow was observable from 0 to <10 μs and then OLED driving current flow stayed unchanged while EL response was gradually decreased.

In addition, we observed no EL with turn the OLED off, while most of typical OLEDs show appreciable EL even turn-off condition due to considerable charge accumulation⁷. Following figure S6 with the enlarged parts at the end of transient EL profiles of both CBP and BSBCz hosts clearly show no such off-state luminescence. This information was added into the manuscript (page 13) and Supplementary Figure S6.

Revision:

In addition, as shown in Supplementary Figure S10, we observed no off-state EL which is a typical observation when there is considerable charge accumulation in OLEDs⁸. Supplementary Figure S10b,d showing the expanded parts at the end of transient EL profiles of both CBP host (**Device A**) and BSBCz host (**Device C**) OLEDs clearly shows no such off-state EL. This result provides additional evidence that charge balance in all devices is ideal and no serious charge accumulation and therefore no SPA occurred in the devices.

Ref. 8: Zhao, D. & Loebel, H. P. The accumulation of diffusive charges in organic light-emitting diodes. *Org. Electron.* **24**, 147–152 (2015).

Supplementary Figure S10. Transient EL responses of OLEDs. (a,b) With the host of CBP for device A and (c,d) with the host of BSBCz for device C. Different current densities with a pulse width of 50 μs were applied to these devices. In the expanded figures (b,d), both devices with CBP and BSBCz hosts showed no off-state EL, indicating that no serious charge accumulation occurred in both devices.

In addition, T. D. Anthopoulos *et al.* reported a highly efficient single layer OLED using CBP as the host with optimized small charge injection layers. Following text (page 10) and reference were added in the manuscript.

Revisions:

T. D. Anthopoulos *et al.* reported a highly efficient single layer OLED device using CBP as the host with optimized small charge injection layers⁴⁹.

Ref. 49: Anthopoulos, T. D. *et al.* Highly efficient single-layer dendrimer light-emitting diodes with balanced charge transport. *Appl. Phys. Lett.* **82**, 4824–4826 (2003).

Comment 3:

3. In Figure 2, the electron-only and hole-only devices were measured for the

DCNP:BSBCz-based single layer devices. It will be useful to compare the difference of unbalanced hole and electron transport for the DCNP:CBP devices.

Response:

We have already answered the detail explanation for this comment at Comment #2.

Comment 4:

4. The transient EL measurements of two host-guest systems were performed to compared the transient EL performances as shown in Figure 4. The authors conclude that the EL intensity drop of the CBP-based device (Device A) is due to the STA effect. However, it is known that the CBP has no good bipolar transport properties. How to exclude the charge-induced quenching effect at high current densities? In addition, it is not clear what is the wavelength for the transient EL measurements in Figure 4 and Supplementary Figure S4.

Response:

- a) We thank for pointing out that the CBP has no good bipolar transport properties and may have charge-induced quenching effects. We have already included the detail explanation in Comment #2.
- b) The transient EL properties were measured using a PMT which collects all visible range emission. The experimental methods and details are given in page 14 and supplementary figure S3. In order to avoid confusion, we added the details into the caption of figure 4 as follows.

Revisions:

The EL intensity from OLEDs was recorded using a photomultiplier tube (PMT) (C9525-02, Hamamatsu Photonics) for the measurement of temporal emission profile.

Comment 5:

5. There are some spelling and grammar errors in the text, which should be corrected.

Response:

We checked the grammar and spelling errors and corrected.

Reviewers' Comments:

Reviewer #2:

Remarks to the Author:

The authors have modified properly the manuscript and the review comments also satisfactory to the acceptance level. After modification the paper quality has been improved and good enough for accepting in this journal. But Before acceptance kindly do some small changes and update the manuscript.

1. Throughout the paper in many places for reference about SI showing "Error! Reference source not found." kindly check and update it properly.

Reviewer #3:

Remarks to the Author:

The authors have fully and carefully addressed the previous comments from the reviewers. The responses are convincing, and the quality of the revised manuscript is highly improved. So, I have no

Response to reviewers

Comments and response (Reviewer #2)

Comment - Remarks to the Author:

The authors have modified properly the manuscript and the review comments also satisfactory to the acceptance level. After modification the paper quality has been improved and good enough for accepting in this journal. But Before acceptance kindly do some small changes and update the manuscript.

Reply:

We are grateful to the reviewer 2 for his/her positive and constructive comments.

Comment 1:

1. Throughout the paper in many places for reference about SI showing “Error! Reference source not found.” kindly check and update it properly.

Response:

We apologize for this mistake and have updated the cross-reference about SI.

Comments and response (Reviewer #3)

Comment - Remarks to the Author:

The authors have fully and carefully addressed the previous comments from the reviewers. The responses are convincing, and the quality of the revised manuscript is highly improved. So, I have no further questions. I strongly recommend the publication of this paper in Nature Communications as is.

Reply:

We are very thankful to the reviewer for his/her recommendation of this paper.